

# Alteration of bacterial community composition in the sediments of an urban artificial river caused by sewage discharge

Yishi Li[1,2,*], Daoming Lou[3,*], Xiaofei Zhou[1], Xuchao Zhuang[2] and Chuandong Wang[1]

[1] State Key Laboratory of Microbial Technology, Microbial Technology Institute, Shandong University, Qingdao, Shandong, China
[2] Focused Photonics (Hangzhou), Inc., Hangzhou, Zhejiang, China
[3] Hangzhou Urban Water Facilities and River Conservation Management Center, Hangzhou, Zhejiang, China
* These authors contributed equally to this work.

Corresponding author
Chuandong Wang,
wangchuandong@sdu.edu.cn

## ABSTRACT

**Background:** Urbanization has an ecological and evolutionary effect on urban microorganisms. Microorganisms are fundamental to ecosystem functions, such as global biogeochemical cycles, biodegradation and biotransformation of pollutants, and restoration and maintenance of ecosystems. Changes in microbial communities can disrupt these essential processes, leading to imbalances within ecosystems. Studying the impact of human activities on urban microbes is critical to protecting the environment, human health, and overall urban sustainability.

**Methods:** In this study, bacterial communities in the sediments of an urban artificial river were profiled by sequencing the 16S rRNA V3-V4 region. The samples collected from the eastern side of the Jiusha River were designated as the JHE group and were marked by persistent urban sewage discharges. The samples collected on the western side of the Jiusha River were categorized as the JHW group for comparative analysis.

**Results:** The calculated alpha diversity indices indicated that the bacterial community in the JHW group exhibited greater species diversity and evenness than that of the JHE group. Proteobacteria was the most dominant phylum between the two groups, followed by Bacteroidota. The relative abundance of Proteobacteria and Bacteroidota accumulated in the JHE group was higher than in the JHW group. Therefore, the estimated biomarkers in the JHE group were divided evenly between Proteobacteria and Bacteroidota, whereas the biomarkers in the JHW group mainly belonged to Proteobacteria. The *Sulfuricurvum*, *MND1*, and *Thiobacillus* genus were the major contributors to differences between the two groups. In contrast to JHW, JHE exhibited higher enzyme abundances related to hydrolases, oxidoreductases, and transferases, along with a prevalence of pathways associated with carbohydrate, energy, and amino acid metabolisms. Our study highlights the impact of human-induced water pollution on microorganisms in urban environments.

## INTRODUCTION

Human settlements have historically and strategically been located near rivers due to the availability of water for subsistence, agriculture, and navigation. The courses of rivers have guided the patterns of human migration and settlement, which have fostered proximity to these essential water sources (*Fang & Jawitz, 2019*). Presently, many densely populated urban centers around the world are geographically positioned in the vicinity of major river estuaries and deltaic regions (*Wijesiri et al., 2019*). These city-affiliated watercourses assume a pivotal role as principal aquatic resources and cater to a spectrum of functions spanning from potable water provisioning to ecosystem services (*Calapez et al., 2023*; *Ferreira et al., 2023*), which influence the overall habitability and quality of life within metropolitan areas (*Tan et al., 2020*). However, with the continuous expansion of cities and acceleration of urbanization, the discharge of various pollutants into urban runoff continues to increase, causing major impacts on ecological sustainability and public health (*Chen et al., 2022*; *Pang, Gao & Guan, 2023*). Water pollution resulting from human activities in urban areas has become a critical environmental concern worldwide (*Wang et al., 2022*). Therefore, understanding the sources, pathways, and impacts of water pollution in urban settings is essential for devising effective mitigation strategies and achieving the sustainability of urban aquatic environments (*Akhtar et al., 2021*; *Muller et al., 2020*).

Urbanization represents an ecosystem evolution mechanism that is intricately influenced by human interventions (*Bruno et al., 2022*). A prominent outcome of this progression are the changes incurred upon the microbiome within aquatic environments, which raise enduring health concerns for aquatic life and human populations (*Zhu et al., 2022*). Because of their roles as producers, consumers, and decomposers, microorganisms are critical components of aquatic ecosystems and key players in maintaining aquatic ecosystem stability, particularly in substance cycling, energy exchange, and pollutant degradation (*Lemaire, Mejean & Iobbi-Nivol, 2020*; *Mishra et al., 2022*; *Peiffer et al., 2021*). However, the precise extent to which urbanization drives microbiota responses in urban waters remains considerably understudied. Microbiome analysis of urban waters has the potential to offer valuable insights into microbes associated with pollution consequences, manifestation of human health hazards, and the degree of human impacts on these freshwater systems (*McLellan, Fisher & Newton, 2015*).

The Jiusha River is an artificial river located in Jianggan District, Hangzhou, Zhejiang Province, China. Jianggan District is one of the five main urban areas of Hangzhou with a floating population of about 1.06 million. It ranks first among Hangzhou's main urban areas and boasts the largest train station and car hub in Hangzhou, along with various traffic elements such as highway junctions and bridges across the river. The total length of the river channel is approximately 5,275 m, with a width of about 30 m and an average depth of around 3.3 m. The water area covers approximately 158,250 square m. Flowing from the east and west sides towards the center, the river eventually converges into Hemu Harbor. The water in the western section of Jiusha River (JHW) appears clearer, whereas the water in the eastern section (JHE) is comparatively turbid. This disparity might be

attributed to the heightened contamination inputs from upstream sources and urban sewage discharge along the route, which have resulted in a discernible degradation in the overall aquatic environment of the eastern section.

This study aims to assess the impact of water pollution resulting from human activities in urban environments on the microbial ecosystem of Jiusha River by comparing the differences in bacterial community structures that colonized in sediment samples collected from the eastern and western sections of this river.

## MATERIALS AND METHODS

### Sample collection and genomic DNA extraction

Sediment samples were collected using grab buckets at five sampling points in both the western and eastern sections of the Jiusha River in July 2023. The sampling points within each group were spaced approximately 500 m apart. Sediment samples were immediately sealed in sterile plastic bags and transported to the laboratory on dry ice and stored at −80 °C until DNA extraction. Genomic DNA was extracted using the Magnetic Soil and Stool DNA Kit (TIANGEN Biotech Co., Ltd., Beijing, China) following the manufacturer's protocol. The purity and concentration of DNA were monitored on 1% agarose gels and evaluated using the ND-1000 NanoDrop spectrophotometer (Thermo Fisher Scientific Inc., Waltham, MA, USA). According to the concentration, DNA was diluted to 1 ng/μL using sterile water.

### Physicochemical analysis of sampling sites

Turbidity was measured using a portable turbidity meter (2100Q; Hach Company, Loveland, CO, USA) and expressed in nephelometric turbidity units (NTUs). Oxidation reduction potential, dissolved oxygen, ammoniacal nitrogen, permanganate index, total phosphorus, and total nitrogen were measured by Zhejiang Hangbang Testing Technology Co., Ltd. pH was analyzed using a PB-10 pH meter (Sartorius, Göttingen, Germany). The C/N ratio was calculated based on carbon and nitrogen content in the sample determined by a micro elemental analyzer (UNICUBE, Elementar, Langenselbold, Germany). The metal concentrations were detected *via* inductively coupled plasma mass spectrometry (NexION-1000G; PerkinElmer, Waltham, MA, USA).

### Gene amplicons and Ilumina Miseq sequencing

The V3–V4 region of the 16S rRNA were targeted for amplification using specific primers 338F/806R with barcodes (*Angebault et al., 2020*; *Zhang et al., 2021*). PCR amplification was performed with Phanta® Max Super-Fidelity DNA polymerase (Vazyme, Nanjing, China) as described previously (*Liu, Zhuang & Wang, 2021*). Sequencing libraries were generated using NEB Next® Ultra™ II FS DNA PCR-free Library Prep Kit (New England Biolabs, Ipswich, MA, USA) and quantified by Qubit (Thermo Scientific, Waltham, MA, USA) and Q-PCR. After the library was qualified, paired-end sequencing was performed with a PE250 strategy *via* the NovaSeq6000 platform (Illumina Inc., San Diego, CA, USA) (*Guo et al., 2023*), which was conducted by Novogene Bioinformatics Technology Co., Ltd. (Beijing, China).

## Sequencing data processing

After assigning the paired-end reads to their respective samples using unique barcodes, forward and reverse sequences were truncated by removing the barcode and primer sequence. Subsequently, the truncated reads were merged by overlapping them using FLASH (version 1.2.11) (*Magoc & Salzberg, 2011*). Quality filtering on the raw tags were performed using the fastp software (version 0.23.1) to obtain high-quality clean tags (*Chen et al., 2018*). The clean tags were compared with the Silva database (https://www.arb-silva.de/) (*Quast et al., 2013*) using the UCHIME algorithm (version 4.2.40) (*Edgar et al., 2011*) to detect and remove off chimera sequences. Denoise was performed with DADA2 plugins in QIIME2 software (version QIIME2-202202) to obtain initial amplicon sequence variants (ASVs) (*Bolyen et al., 2019*). Species annotation was performed using QIIME2 software with default parameters based on the Silva database for 16S regions. Based on the sequencing data, PICRUSt2 (https://github.com/picrust/picrust2) (*Douglas et al., 2020*) was used for predictive functional analysis of bacterial communities.

## Statistical analysis

Alpha diversity indices (Chao1, Dominance, Observed species, Pielou E, Shannon, Simpson) were calculated by QIIME2 to show the species complexity of collected samples (*Xia & Sun, 2023*). Non-metric multidimensional scaling (NMDS) analysis based on the Bray-Curtis dissimilarity matrices was performed to visualize broad trends of similarities and differences of related samples by ggplot2 packages in R software (version 4.0.3) (*R Core Team, 2020*; *Villanueva & Chen, 2019*). A cluster tree based on the weighted unifrac distance matrix was constructed using the unweighted pair group method with arithmetic mean (UPGMA) algorithm, and the UPGMA diagram was drawn within QIIME2. Biologically relevant features were performed using linear discriminant analysis effect size (LEfSe) (*Segata et al., 2011*). One-way analysis of variance (ANOVA) was performed with SPSS software (version 22.0; SPSS Inc., Chicago, IL, USA).

# RESULTS

## Comparison of the physicochemical properties and bacterial community diversity in collected samples

Ten sampling points were established on both the eastern and western sides of the Jiusha River, which were categorized into the JHE and JHW groups (Fig. 1A). Subsequent water quality assessments at these sampling sites revealed notable distinctions (Table 1). It was observed that the water from the JHW group exhibited higher clarity with turbidity levels ranging from 35 to 39 NTUs, whereas the JHE group exhibited significantly ($p = 0.0002$) higher turbidity levels, ranging from 42 to 47 NTUs. Additionally, the JHW group demonstrated lower concentrations of ammoniacal nitrogen (AN) (0.63 to 0.75 mg/L) and total phosphorus (TP) (0.18 to 0.33 mg/L), while these two parameters were notably higher ($p = 0.0005$ for AN and 0.0327 for TP) in the JHE group, ranging from 1.15 to 1.90 mg/L and 0.3 to 0.39 mg/L, respectively. Additionally, the pH values of the JHW group consistently exceeded 8, whereas at JHE1-3 sites within the JHE group, pH values declined to a range of 7.4–7.7. Concurrently, these three locations also displayed the highest values
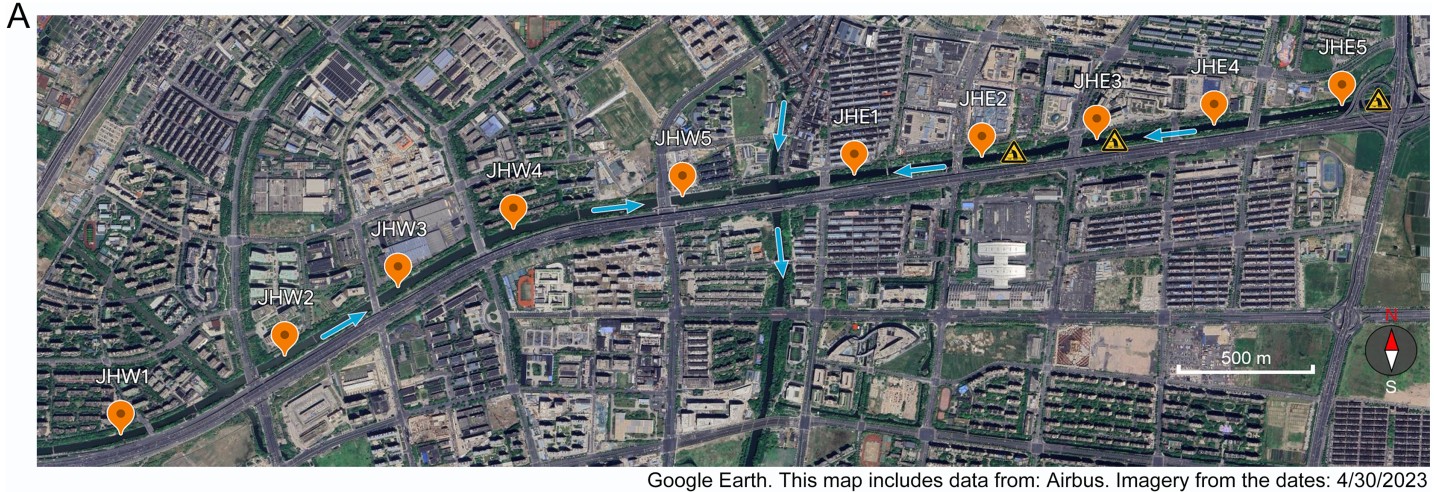

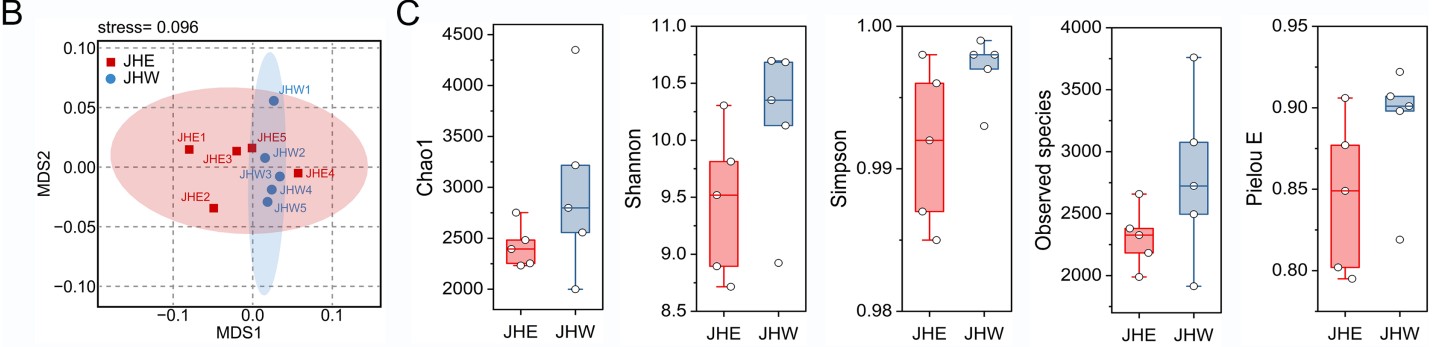

**Figure 1** **The geographical and biological characteristics of the sampled sites.** (A) Location distribution map of sampling points on the eastern (JHE1-5) and western (JHW1-5) sides of Jiusha River. The blue arrows indicate the direction of river water flow. The yellow triangular symbols in the diagram represent the presence of sewage discharge outlets at these locations. The bar represents 500 m. (B) The similarity of sampling points based on the physicochemical properties and partial metal content listed in Table 1 as demonstrated by NMDS analysis. (C) Comparison analysis of alpha diversity indices between the eastern (JHE) and western (JHW) sides of the Jiusha River. The boxplots show median values and interquartile ranges. Map includes data from: Airbus, Imagery from 4/30/2023.

of oxidation reduction potential (ORP) and the lowest dissolved oxygen (DO) levels. Next, the contents of common elements in the sediment samples at each collection point were detected. Although there was no significant difference ($p = 0.5314$) in C/N ratio between the JHW and JHE groups, the C/N ratio in the JHW group exhibited relatively low variability (ranging from 12.18 to 19.87), while the C/N ratio in the JHE group displayed a wider range of fluctuations (from 5.45 to 36.68). According to the NMDS analysis results, it was evident that the samples within the JHW group were closely clustered together, whereas those in the JHE group were more widely dispersed (Fig. 1B). This indicated that the physicochemical characteristics of the collection points within the JHW group were more similar than those in the JHE group. A total number of 780,751 paired-end reads were originally obtained from the sequencing instrument (Table S1). After filtering for low quality, short length, and chimeric sequences, 492,777 effective tags were finally obtained with >50% GC content, and Q20 and Q30 values of >95%. The calculated mean values of Chao1, Shannon, Simpson, and Pielou E indices for the JHW group were all higher than those of the JHE group, indicating that the bacterial community in the JHW group

**Table 1  The location, physicochemical properties, and partial metal contents of the sampling sites.**

| | JHE1 | JHE2 | JHE3 | JHE4 | JHE5 | Mean | Stderr | JHW1 | JHW2 | JHW3 | JHW4 | JHW5 | Mean | Stderr | $p$ value |
|---|---|---|---|---|---|---|---|---|---|---|---|---|---|---|---|
| Coordinates | 30°19′03″N 120°16′07″E | 30°19′05″N 120°16′23″E | 30°19′07″N 120°16′39″E | 30°19′09″N 120°16′54″E | 30°19′11″N 120°17′12″E | | | 30°18′32″N 120°14′29″E | 30°18′41″N 120°14′51″E | 30°18′49″N 120°15′06″E | 30°18′56″N 120°15′21″E | 30°19′00″N 120°15′44″E | | | |
| **Supernatant** | | | | | | | | | | | | | | | |
| pH | 7.4 | 7.7 | 7.5 | 8.2 | 8.4 | 7.84 | 0.196 | 8.5 | 8.3 | 8.1 | 8.2 | 8.4 | 8.3 | 0.071 | 0.0587 |
| Temperature (°C) | 28.9 | 28.8 | 28.8 | 28.6 | 28.9 | 28.8 | 0.054 | 29.3 | 29.5 | 29.4 | 29.6 | 29.5 | 29.46 | 0.051 | 2.1E−5 (***) |
| ORP (mV) | 365 | 341 | 318 | 301 | 299 | 324.8 | 12.56 | 307 | 301 | 298 | 306 | 308 | 304 | 1.924 | 0.1404 |
| DO (mg/L) | 4.55 | 3.83 | 4.52 | 6.89 | 6.97 | 5.352 | 0.657 | 5.53 | 6.78 | 6.33 | 6.41 | 6.47 | 6.30 | 0.208 | 0.2045 |
| Turbidity (NTUs) | 43 | 45 | 42 | 47 | 44 | 44.2 | 0.860 | 38 | 35 | 37 | 36 | 39 | 37 | 0.707 | 0.0002 (***) |
| AN (mg/L) | 1.27 | 1.15 | 1.89 | 1.64 | 1.90 | 1.57 | 0.155 | 0.75 | 0.63 | 0.72 | 0.67 | 0.66 | 0.686 | 0.022 | 0.0005 (***) |
| PI (mg/L) | 5.9 | 6.0 | 5.0 | 5.3 | 5.5 | 5.54 | 0.186 | 4.3 | 3.4 | 4.1 | 7.3 | 7.0 | 5.2 | 0.803 | 0.7081 |
| TP (mg/L) | 0.39 | 0.35 | 0.3 | 0.39 | 0.3 | 0.346 | 0.020 | 0.22 | 0.18 | 0.2 | 0.32 | 0.33 | 0.25 | 0.031 | 0.0327 (*) |
| TN (mg/L) | 2.97 | 3.19 | 3.5 | 3.08 | 3.24 | 3.196 | 0.089 | 2.7 | 2.53 | 2.3 | 4.41 | 4.12 | 3.212 | 0.437 | 0.9723 |
| **Sediment** | | | | | | | | | | | | | | | |
| Depth (m) | 7 | 8 | 8 | 9 | 8 | 8 | 0.316 | 7 | 8 | 7 | 7 | 8 | 7.4 | 0.244 | 0.1720 |
| C/N ratio | 5.45 | 11.84 | 22.09 | 36.68 | 22.64 | 19.74 | 5.327 | 14.52 | 12.18 | 16.88 | 17.31 | 19.87 | 16.152 | 1.306 | 0.5314 |
| Al (mg/kg) | 0.37 | 0.28 | 0.12 | 0.21 | 0.07 | 0.21 | 0.054 | 0.24 | 0.28 | 0.12 | 0.15 | 0.14 | 0.186 | 0.031 | 0.7103 |
| Ca (mg/kg) | 12.64 | 11.11 | 13.04 | 5.86 | 10.02 | 10.53 | 1.288 | 7.66 | 14.34 | 9.29 | 8.21 | 10.16 | 9.932 | 1.184 | 0.7396 |
| Fe (mg/kg) | 0.12 | 0.21 | 0.12 | 0.19 | 0.16 | 0.16 | 0.018 | 0.32 | 0.14 | 0.27 | 0.22 | 0.14 | 0.218 | 0.036 | 0.1844 |
| K (mg/kg) | 21.24 | 19.38 | 19.02 | 10.65 | 20.30 | 18.12 | 1.906 | 12.95 | 19.53 | 17.34 | 11.11 | 12.99 | 14.78 | 1.567 | 0.2137 |
| Mg (mg/kg) | 12.28 | 7.91 | 11.78 | 4.48 | 8.33 | 8.956 | 1.424 | 17.54 | 11.57 | 12.75 | 4.31 | 4.49 | 10.13 | 2.545 | 0.6973 |
| Na (mg/kg) | 64.22 | 29.67 | 59.14 | 43.84 | 57.85 | 50.94 | 6.301 | 84.95 | 49.35 | 34.13 | 42.41 | 31.13 | 48.39 | 9.681 | 0.8308 |

**Notes:**
*** $p < 0.001$.
* $p < 0.05$.
Stderr, standard error.
ORP, oxidation reduction potential; DO, dissolved oxygen; NTUs, nephelometric turbidity units; AN, ammoniacal nitrogen; PI, permanganate index; TP, total phosphorus; TN, total nitrogen; C/N ratio, carbon to nitrogen ratio.

exhibited greater species diversity and evenness (Fig. 1C). The highest number of observed species was in JHW5 (3,760), followed by JHW3 (3,075), and the lowest was in JHE1 (1,989) and JHW2 (1,914) (Table S2).

## Differential analysis of dominant bacterial distribution between the JHW and JHE groups

To assess the degree of similarity among the samples, a clustering tree of the samples was constructed, and the clustering result was integrated with the relative species abundance of each sample at the phylum level for presentation (Fig. 2A). All samples were divided into two main clusters, with samples from the JHW group and the JHE group falling neatly into these respective clusters, which indicated that the phylogenetic relationship of the JHW group was relatively far from the JHE group. At the phylum level, the top 10 most dominant phyla in the samples were Proteobacteria (22.76~60.77%), Bacteroidota

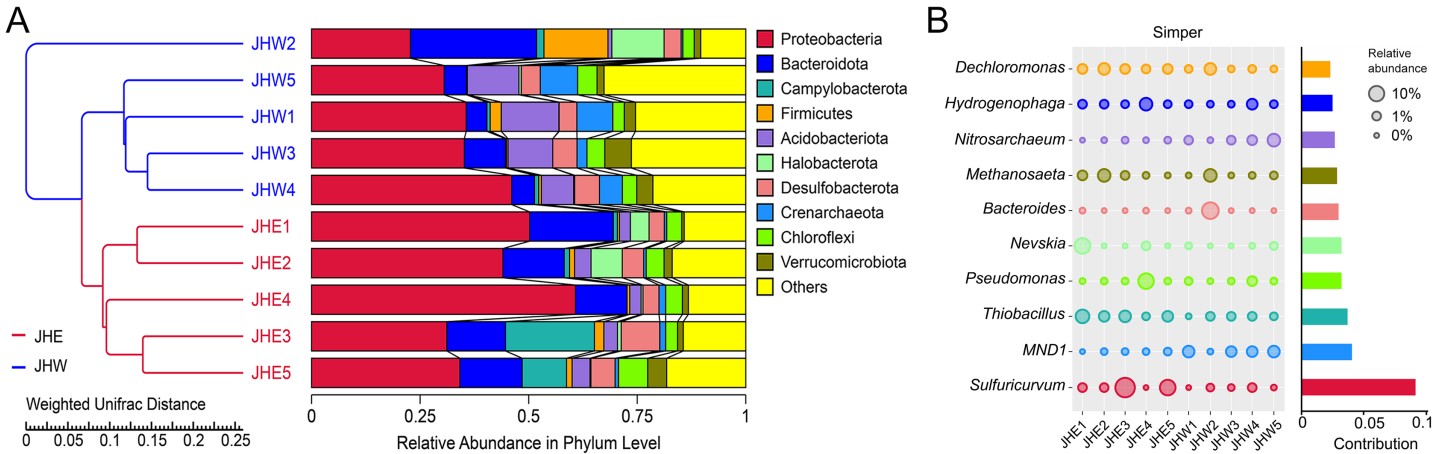

**Figure 2 Distribution of species abundance in collected samples by phylum and genus classification.** (A) UPGMA tree based on weighted UniFrac distance at the phylum level. The left side is the UPGMA clustering tree structure, and the right side is the relative abundance distribution of the top 10 most abundant species in each sample at the phylum level. The remaining phyla are grouped as others. (B) Top 10 bacterial genera contributing most to the differences between the JHE and JHW groups based on similarity of percentages (Simper) analysis. The size of the bubble chart on the left represents the relative abundance of the species, and the length of the histogram on the right represents the contribution of the species to the difference between the two groups.

(4.77~29.09%), Campylobacterota (0.00~20.45%), Firmicutes (0.14~14.74%), Acidobacteriota (0.89~13.26%), Halobacterota (0.02~12.04%), Desulfobacterota (3.51~8.88%), Crenarchaeota (0.40~8.60%), Chloroflexi (2.64~6.75%), and Verrucomicrobiota (0.57~6.12%) (Table S3). Among them, the most abundant Proteobacteria were mainly detected in JHE4 (60.77%) and JHE1 (50.20%), followed by JHW4 (46.08%) and JHE2 (44.08%). Compared to others, Proteobacteria dominated in both groups, while Acidobacteriota and Crenarchaeota were predominantly distributed in the JHW group. To determine the genera mainly responsible for the differences in bacterial communities between the JHE group and JHW group, the contribution of each species to the differences between the two groups was quantified based on Simper analysis. It indicated that the major dissimilarity contributors were *Sulfuricurvum*, *MND1*, and *Thiobacillus* (Fig. 2B). *Sulfuricurvum* that accounted for the majority of variations in the community structure demonstrated its highest relative abundance in JHE3 (19.35%) and JHE5 (9.52%), but was undetected in both JHE4 and JHW1 (Table S4). The relative abundance of *MND1* is greater in the JHW group (0.01~3.85%) than in the JHE group (0.00~0.54%), while *Thiobacillus* was predominantly found in the JHE group (0.59~5.94%) compared to the JHW group (0.02~1.13%).

An LEfSe analysis was subsequently performed to compare the estimated biomarkers between the two groups. The Hydrogenophilaceae, Bacteroidetes vadinHA17, Prolixibacteraceae, Lentimicrobiaceae, Methylophilaceae, Halomonadaceae, and Bacteroidetes BD2-2 family; the *Thiobacillus*, *Nitrosomonas*, *Sulfurisoma*, *Curvibacter*, *Anaerolinea*, *Cavicella*, *unidentified Hydrogenophilaceae*, and *Halomonas* genus; and the *Candidatus Halomonas phosphatis* species were enriched in the JHE group (Fig. 3A). These enriched phylotypes were predominantly from the Bacteroidetes BD2-2, Bacteroidetes vadinHA17, Prolixibacteraceae, Lentimicrobiaceae, Hydrogenophilaceae,

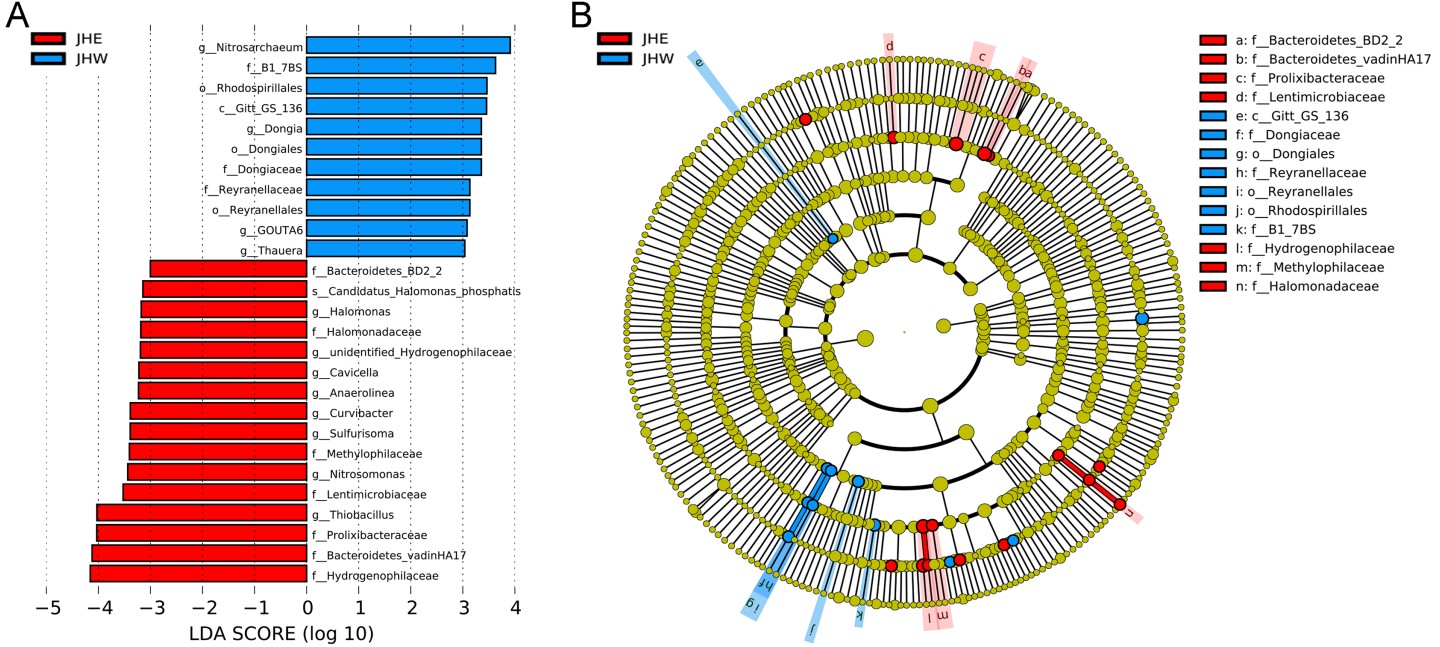

**Figure 3 Key phylotypes of differently abundant taxa identified between the JHE and JHW groups using linear discriminant analysis (LDA) combined with effect size (LEfSe) algorithm.** (A) Histogram of the LDA scores (LDA score > 3, *p* value > 0.05) computed for differentially abundant bacterial taxa between the two groups. (B) Cladogram generated using the LEfSe method illustrating the significant phylogenetic distribution of bacteria associated with the JHE group (red) and JHW group (blue). Non-significant discriminant taxonomic nodes are colored yellow. The concentric circles radiating outward represent classification levels from phylum to species. The diameter of the circles is directly proportional to the relative abundance.                                                           

Methylophilaceae, and Halomonadaceae family (Fig. 3B). In contrast, the Gitt-GS-136 class, the Rhodospirillales, Dongiales, and Reyranellales order; the B1-7BS, Dongiaceae, and Reyranellaceae family, the *Nitrosarchaeum*, *Dongia*, *GOUTA6*; and *Thauera* genus were enriched in the JHW group (Fig. 3A). These enriched phylotypes were predominantly from the Gitt-GS-136 class and the Dongiales, Reyranellales, and Rhodospirillales order, as well as the Dongiaceae, Reyranellaceae, and B1-7BS family (Fig. 3B).

## Comparative analysis of functional disparities within bacterial communities

In total, 22 enzyme commission (EC) numbers predicted by PICRUSt2 exhibited significant differences (*p* value < 0.01) between the JHW and JHE groups (Fig. 4A). These EC numbers were predominantly enriched in six functional enzyme categories, namely ligases, lyases, oxidoreductases, transferases, hydrolases, and isomerases. After comparing the differences between groups, we observed that the average proportion of 13 EC numbers in the JHE group were significantly higher than others in the JHW group. Among these, the highest mean proportion was correlated to the member of isomerases (EC:5.2.1.8), with a relative abundance of 0.930 ± 0.021% in the JHE group compared to 0.863 ± 0.025% in the JHW group. This was closely followed by EC:3.6.4.13 belonging to the hydrolases, with a mean proportion of 0.408 ± 0.013% in the JHE group and 0.357 ± 0.018% in the JHW group. Conversely, nine EC numbers had significantly higher mean proportions in the

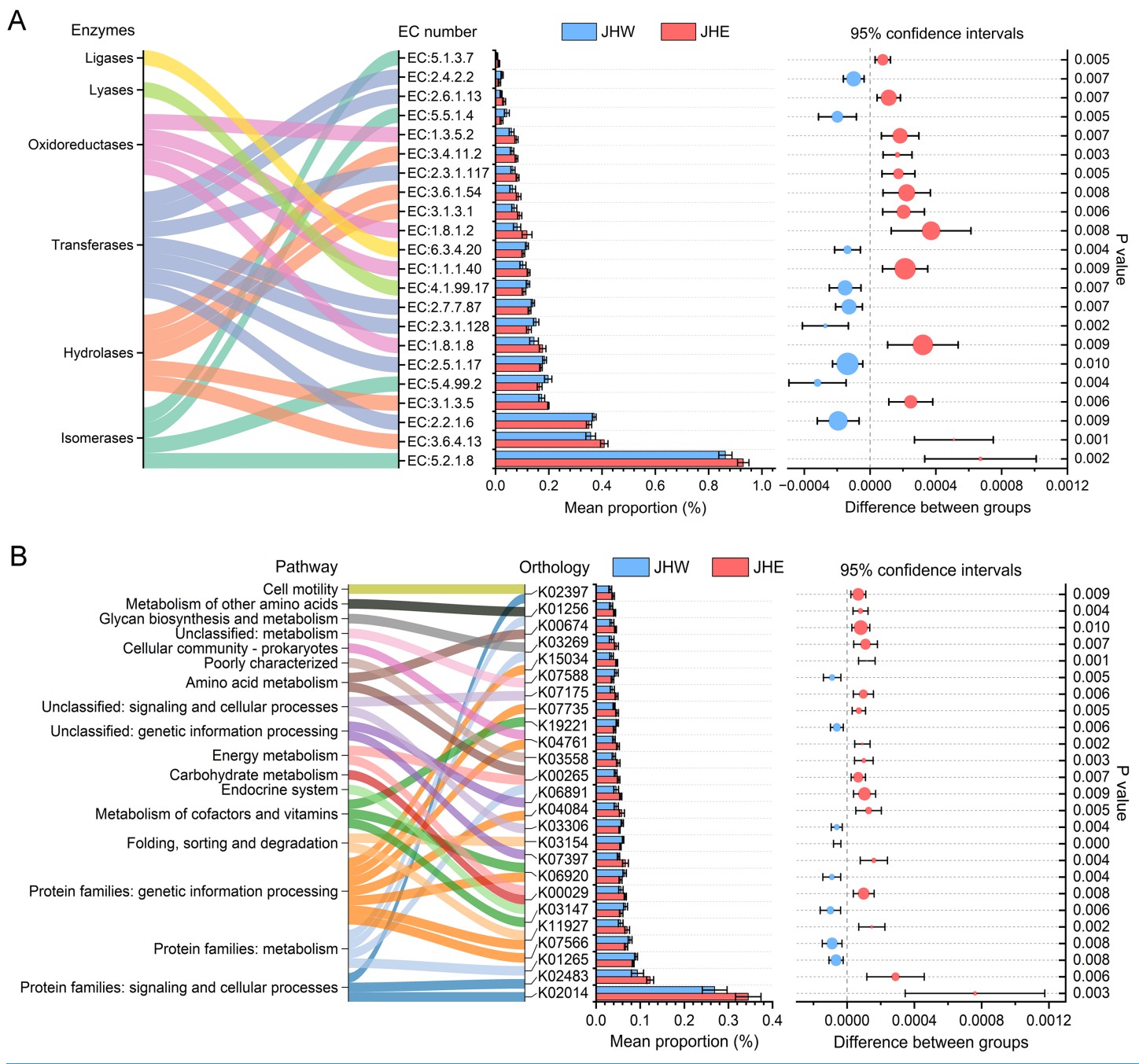

**Figure 4 Differences in the abundances of enzymes (A) and pathways (B) inferred by PICRUSt2.** The bar plot shows mean proportions (%) of differential EC numbers and KEGG orthologs between the JHE and JHW groups. Sankey diagram on the left illustrates the enriched enzymes and metabolic pathways. The differences in proportions between groups are depicted with 95% confidence intervals on the right, and only Bonferroni-adjusted *p*-values ≤ 0.01 are shown.

JHW group than in the JHE group. For instance, EC:2.2.1.6 (a transferase) exhibited mean proportions of 0.351 ± 0.001% in the JHE group and 0.370 ± 0.007% in the JHW group. The most enriched enzyme class was transferase, which contained a total of seven EC numbers. Except for EC:2.3.1.117 and EC:2.6.1.13, which were higher in the JHE group,

the remaining five EC numbers within this class were more abundant in the JHW group. The second most enriched enzymes were hydrolases, which consisted of five EC numbers, all of which exhibited higher abundance in the JHE group. Additionally, the enriched oxidoreductases also demonstrated greater prevalence in the JHE group. Conversely, the least enriched enzymes were ligases and lyases, which were both represented by a single EC number and were more abundant in the JHW group.

In addition to the predicted enzyme functions, 25 KEGG orthologs belonging to 17 different pathways also exhibited significant differences between the JHE and JHW groups (Fig. 4B). A total of 68% (17/25) of the predicted KEGG orthologs showed higher mean proportions in the JHE group than the JHW group. The highest mean proportion was associated with protein families related to signaling and cellular processes (K02014), with a mean proportion of $0.345 \pm 0.029\%$ in the JHE group compared to $0.269 \pm 0.028\%$ in the JHW group. The most enriched pathway were the protein families related to genetic information processing, which contained a total of seven KEGG orthologs. The abundance of metabolism-related pathways in the JHE group were generally higher than those in the JHW group, such as carbohydrate metabolism, energy metabolism, amino acid metabolism, glycan biosynthesis and metabolism, and metabolism of other amino acids.

## DISCUSSION

Morphological and hydrological changes to water bodies are common in urban landscapes due to high anthropogenic forcing, which often results in poor water quality in these systems and has significant impacts on the provision of ecosystem services (*Teurlincx et al., 2019*). AN and TP are widely used as indicators in the assessment and monitoring of water quality status (*Yang et al., 2021*). High concentrations of AN and TP frequently resulted in eutrophication, excessive oxygen consumption, and production of blackening and stinking pollutants, which ultimately cause discoloration and unpleasant odor in rivers, which are particularly pronounced during the summer season (*Dębska et al., 2021*; *Yu et al., 2020*). Due to continuous discharge of urban sewage along its course, the JHE group exhibited significantly higher levels of AN and TP compared to the JHW group (Table 1), which then contributed to the relatively turbid water observed in the JHE group. Although no significant difference in pH was observed between the JHE and JHW groups, a decreasing trend in pH along the course of the river was still detected in the JHE group, which dropped from 8.4 at JHE5 to 7.4 at JHE1. This trend was also attributed to the influence of coastal sewage discharge, as the pH levels in the control JHW group were consistently maintained between 8.1 and 8.5. The acidification of surface water due to acidic deposition has garnered widespread concern as a critical environmental issue (*Grennfelt et al., 2020*). While surface water acidification might not currently be considered a serious regional issue in China, the increased potential for acidification in susceptible water bodies suggested that ignoring the risks of urban surface water acidification was not advisable, both in the present and future (*Qiao et al., 2016*; *Yu et al., 2017*). The varied C/N ratio of sediments was suggested to indicate a progressive change in sources of organic matter and depositional environment (*Venkatesh & Anshumali, 2020*). Lower C/N ratios were due to a shift from nitrogen-limited to nitrogen-rich bed sediment, while the deposition of organic

matter through runoff resulted in higher C/N ratios. As organic pollution from upstream sources increased and urban sewage discharged along the route, urban river sediments accumulated a significant load of anthropogenic carbon and nitrogen, which possibly resulted in notable fluctuations in the C/N ratios within the JHE group, which varied from 5.45 in JHE1 to 36.68 in JHE4. In contrast, the C/N ratios of the unaffected JHW group remained relatively stable. ANOVA indicated no significant difference in the C/N ratios between the JHE and JHW groups, which might be attributed to the larger standard error induced by the overall larger variability in C/N ratios within the JHE group.

Most aquatic organisms are very sensitive to changes in their environment and respond to pollution in a variety of ways that can ultimately lead to the loss of biodiversity in the polluted local area, such as death or migration to another environment (*McDonald et al., 2020*; *Ogidi & Akpan, 2022*). Similar to macroorganisms, microbial communities in urban aquatic ecosystems are also susceptibly influenced by spatial variation in physicochemical parameters, which can be used as bioindicators of environmental quality (*Sagova-Mareckova et al., 2021*). Numerous previous studies have repeatedly reported a reduction in microbial taxonomic diversity within polluted urban water bodies (*Ibekwe, Ma & Murinda, 2016*; *Paruch et al., 2019*; *Wang et al., 2021*). Our findings aligned with this observation, as we found that several alpha diversity indices were consistently lower in the JHE group, which was characterized by higher turbidity and AN and TP content, when compared to the JHW group (Fig. 1C and Table 1). The higher Pielou E values observed in the JHW group suggested a more even distribution of species within the bacterial community compared to the JHE group. This phenomenon could be attributed to the relatively consistent physicochemical parameters found among samples within the JHW group, as depicted in Fig. 1B. Indeed, environmental factors play a decisive role in shaping aquatic microbial communities (*Philippot, Griffiths & Langenheder, 2021*). Therefore, the bacterial community of the JHW group exhibited more similar relative abundances at the phylum level and formed a distinct branch separated by weighted unifrac distance from the JHE group (Fig. 2A).

Despite differences in the relative abundance of Proteobacteria between the JHE and JHW groups, it remained the dominant phylum in all individual samples (Fig. 2A). The *MND1*, *Thiobacillus*, *Pseudomonas*, *Nevskia*, *Hydrogenophaga*, and *Dechloromonas* genus that contributing most to the differences between the JHE and JHW groups were all members of the Proteobacteria phylum (Fig. 2B). This aligns with previous findings in urban river microbial communities affected by anthropogenic activities, where Proteobacteria consistently emerge as the most abundant phylum (*Feng et al., 2022*; *Godoy et al., 2020*). Bacteroidota is another phylum with high relative abundance in the JHE and JHW groups after Proteobacteria (Fig. 2A). An investigation into the bacterial community composition in river sediments from the south of Zhejiang Province found that the abundance of Proteobacteria and Bacteroidetes increased in areas contaminated by wastewater (*Lu et al., 2016*), which supported our findings that the relative abundance of Proteobacteria and Bacteroidota accumulated in the JHE group was higher than the JHW group (Fig. 2A). The higher abundance exhibited by the Proteobacteria and Bacteroidota is considered closely associated with anthropogenic fecal contamination (*Ibekwe, Ma &*

*Murinda, 2016; Paruch et al., 2019; Teixeira et al., 2020*). Firmicutes, Actinobacteria, and Chloroflexi, which were not prevalent in the JHE and JHW groups, were reported to be dominant in the Qiantang River and estuaries of the unpolluted river in Zhejiang Province, and ranked second only to Proteobacteria and Bacteroidetes in terms of abundance (*Liu et al., 2015; Lu et al., 2016*). In the sediments of Zhangxi River, a peri-urban river located in the Zhejiang Province, Proteobacteria and Bacteroidetes were identified as the dominant phyla in densely populated areas, while Proteobacteria and Cyanobacteria were prevalent in sparsely populated regions (*Zheng et al., 2018*). Hence, in contrast to Proteobacteria, which appeared to be the predominant phylum regardless of the level of pollution and population density in the areas where the river flowed, Bacteroidetes were observed in higher proportions in regions with greater anthropogenic influence, at least in the context of Zhejiang Province. This observation was further supported by the findings in this article, which indicated that the estimated biomarkers in the JHE group belonged equally to Proteobacteria and Bacteroidota, whereas the biomarkers in the JHW group mainly belonged to Proteobacteria (Fig. 3).

Proteobacteria encompass the majority of traditional Gram-negative bacteria and demonstrate remarkable metabolic diversity and biological significance (*Kersters et al., 2006*). Increasing data indicate that Proteobacteria are potential microbial markers for various diseases, including metabolic disorders, inflammatory bowel disease, asthma, and chronic obstructive pulmonary disease (*Rizzatti et al., 2017*). Most members of Bacteroidota maintain a complex and generally beneficial relationship with their hosts in the intestinal environment, which metabolize polysaccharides and oligosaccharides to provide nutrients and vitamins to the host and other gut microbial inhabitants (*Zafar & Saier, 2021*). However, when they escape this environment, they can cause severe anaerobic bacterial infections, including bacteremia and abscess formation in multiple body sites (*Wexler, 2007*). Therefore, the risk management of potential bacterial pathogens in urban surface waters should be appropriately managed. Microorganisms are crucial participants in supporting ecosystem functionality, primarily due to their involvement in biogeochemical cycles and intrinsic connection to the resistance and resilience of ecosystems (*Philippot, Griffiths & Langenheder, 2021; Ranheim Sveen, Hannula & Bahram, 2023*). Predicting the functional characteristics of bacterial communities can provide valuable predictive information for exploring microbial regulation of feedback to ecosystem change. Microbial enzymes can achieve effective biodegradation during bioremediation through multipollutant mechanisms such as oxidation, elimination, ring opening, and reduction (*Narayanan, Ali & El-Sheekh, 2023*). Hydrolases, oxidoreductases, transferases, and isomerases represent the major classes of microbial enzymes accountable for the degradation of a wide range of toxic pollutants within the environment and are widely used in bioremediation through immobilization and genetic engineering techniques (*Narayanan, Ali & El-Sheekh, 2023; Saravanan et al., 2021*). The JHE group exhibited a higher abundance of enzymes related to hydrolases, oxidoreductases, and transferases (Fig. 4A), suggesting that bacterial communities within sediments played an important role in the self-purification of aquatic ecosystems and the remediation of external pollution. In this study, metabolism and genetic information processing were observed to

be the two predominant functions of bacteria in sediments (Fig. 4), which was consistent with prior research findings (*Srivastava & Verma, 2023*; *Yang et al., 2022*). Certain taxa can establish intricate mutualistic relationships with other taxa through metabolic pathways involved in amino acid, carbohydrate, cofactor, and vitamin metabolism, which are crucial for maintaining the structure and function of a stable microbiome (*Morris et al., 2013*; *Scherlach & Hertweck, 2018*; *Yang et al., 2023*). This was consistent with our findings, as we predicted the presence of numerous pathways related to cofactors, vitamins, and carbohydrate and amino acid metabolism in the river sediments (Fig. 4B). In addition, the JHE group revealed a higher proportion of pathways associated with carbohydrate, energy, and amino acid metabolism, as well as glycan biosynthesis (Fig. 4B), suggesting that the presence of rich organic compounds facilitated the growth of bacterial communities within this group.

## CONCLUSION

The present study provided insight into the impact of water pollution resulting from human activities on bacterial communities in the sediments of an urban artificial river. The JHW group had better water quality and exhibited greater species diversity and evenness than the JHE group, which was characterized by lower water quality. Both groups were dominated by Proteobacteria and Bacteroidota, but the cumulative abundance of Proteobacteria and Bacteroidetes was higher in JHE. Notably, biomarkers in JHE were evenly distributed between Proteobacteria and Bacteroidota, while those in JHW predominantly belonged to Proteobacteria. The *Sulfuricurvum*, *MND1*, and *Thiobacillus* genera were the major contributors responsible for the differences between the two groups. In contrast to the JHW group, the JHE group exhibited a higher abundance of enzymes related to hydrolases, oxidoreductases, and transferases, as well as a prevalence of pathways associated with carbohydrate, energy, and amino acid metabolism. This research enhances our understanding of the intricate interactions between urbanization and ecosystems, and forms a crucial foundation for informed decision-making and responsible urban development.

## ACKNOWLEDGEMENTS

The authors would like to thank Chengjia Zhang of the Core Facilities for Life and Environmental Sciences, State Key Laboratory of Microbial Technology of Shandong University for the technical assistance in elemental analysis.

### Funding

This work was supported by the Shandong Provincial Natural Science Foundation (No. ZR2021QC087). The funders had no role in study design, data collection and analysis, decision to publish, or preparation of the manuscript.

### Grant Disclosures

The following grant information was disclosed by the authors:
Shandong Provincial Natural Science Foundation: ZR2021QC087.

### Competing Interests

Yishi Li and Xuchao Zhuang are employed by Focused Photonics (Hangzhou), Inc. Daoming Lou is employed by Hangzhou Urban Water Facilities and River Conservation Management Center.

### Author Contributions

- Yishi Li conceived and designed the experiments, performed the experiments, analyzed the data, prepared figures and/or tables, authored or reviewed drafts of the article, and approved the final draft.
- Daoming Lou performed the experiments, analyzed the data, authored or reviewed drafts of the article, and approved the final draft.
- Xiaofei Zhou performed the experiments, prepared figures and/or tables, authored or reviewed drafts of the article, and approved the final draft.
- Xuchao Zhuang performed the experiments, analyzed the data, authored or reviewed drafts of the article, and approved the final draft.
- Chuandong Wang conceived and designed the experiments, analyzed the data, prepared figures and/or tables, authored or reviewed drafts of the article, and approved the final draft.

### Data Availability

The sequencing data are available in the NCBI Sequence Read Archive: PRJNA1019893.

### Supplemental Information

Supplemental information for this article can be found online at http://dx.doi.org/10.7717/peerj.16931#supplemental-information.

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
