# Peer review of "Alteration of bacterial community composition in the sediments of an urban artificial river caused by sewage discharge"

_PeerJ, doi:10.7717/peerj.16931_

## Round 0.1 · original submission · Major Revisions

Please revise the manuscript thoroughly and provide point-by-point response letter when resubmitting your work to the editorial office

**Language Note:** The review process has identified that the English language must be improved. PeerJ can provide language editing services - please contact us at copyediting@peerj.com for pricing (be sure to provide your manuscript number and title). Alternatively, you should make your own arrangements to improve the language quality and provide details in your response letter. – PeerJ Staff

·

Basic reporting

The purpose of this manuscript is to evaluate the effects of human activities on microbial ecosystem of Jiusha River by comparing the difference of bacterial community structure in sediments of Jiusha River in the eastern and western sections of Hangzhou, Zhejiang Province.

Experimental design

The selection of sampling points, sampling methods and statistical analysis are basically reasonable, but the analysis of the results needs to be further refined to provide a basis for hypotheses, guesses and relevant literature supporting possible cause explanations.

Validity of the findings

The results show that the water quality in the western section of Jiusha River is good, while the water quality in the eastern section is poor. The reason for this difference is that upstream pollution sources and urban sewage discharge along the route have increased, resulting in significant degradation of the overall water environment in the eastern section.

Additional comments

The following issues also need to be explained and improved.
1. In 161-164 lines, why is there such a big gap between the C/N ratio of JHE1 and JHE4 groups? Please interpret or cite the literature.
2. In 216-236 lines, the analysis of experimental results only analyzed the changes of microbial community quantity and the comparison of some enzymes quantity on the east and west banks of the Jiusha River, and did not explain the production and change mechanism of each bacterium and enzyme in detail. Please analyze and supplement the scientific nature of the manuscript.
3. In Table 1, ANOVA was not performed on all data, please add.
4. In Table 1, is AI the heavy metal aluminum (Al)? Please check the full manuscript.

Reviewer 2 ·

Basic reporting

no comment

Experimental design

no comment'

Validity of the findings

no comment'

Additional comments

In the manuscript "Alteration of bacterial community composition in the
sediments of an urban artificial river caused by sewage discharge" the authors show the differences in the bacterial community of two sites in an urban artificial river in China. the results show that there are changes between two analyzed bacterial communities, as well as in some physicochemical parameters examined. However, there are some important points to consider in this work.

92 samples were taken throughout the 5,275 meters of the Jiusha River?
Is there any part of the Jiusha River that does not pass through a non-urban area?

It is not clear if the ten sites along the Jiusha River sampled are urban areas. How populated are these areas?

281 In the discussion section, the authors suggest that "The high
abundance exhibited by the Proteobacteria and Bacteroidota is considered closely associated with anthropogenic fecal contamination" and conclude "the relative abundance of Proteobacteria and Bacteroidota accumulated in the contaminated JHE group was higher than the uncontaminated JHW group".
Bacteroidota accumulated in the contaminated JHE group was higher than the uncontaminated JHW group". When looking at Figure 2A, certainly the proportion of Proteobacteria and Bacteroidota in the JHE group is higher than in the JHW group, however also in the JHW group also have Proteobacteria and Bacteroidota, therefore one can conclude that JHW group are also contaminated this zone, possibly less than the JHE group, but not uncontaminated as the authors claim.

307 the authors claim: "The JHE group exhibited a higher abundance of enzymes related to hydrolases, oxidoreductases, and transferases" and subsequently: "the JHE group revealed a prevalence of pathways associated with carbohydrate, energy, and amino acid metabolisms (Fig. 4B), indicating that river sediments were rich in organic compounds, thereby supporting the growth of bacterial communities associated with carbohydrate, energy, and amino acid metabolisms (Fig. 4B), indicating that river sediments were rich in organic compounds, thereby supporting the growth of bacterial communities." It is true that in Figure 4 it is observed that the abundance of enzymes and metabolic pathways in the JHE group is statistically higher than in the JHW group, but also these enzymes and pathways are present in the JHW group, i.e. this group is less contaminated, but it is contaminated.

One aspect that the authors do not discuss in the manuscript are the results of table 1, only mentioned in the part of the results 151-164, these differences in the different physicochemical values analyzed between the two groups how are they interpreted? turbidity, pH, C/N ratio, among others that have to do with water quality? further discussion of these data would be desirable.

The major problem that I observe is that it would have been desirable to compare the data obtained in this manuscript, with data of waters of the same type, from some nearby non-urban place, since Jiusha river is totally urban therefore there is no alteration of the bacterial community, this is the bacterial community that exists in this urban river. As the authors demonstrate, one part is more contaminated than the other, but both are contaminated and as the authors suggest the cause: "can be attributed to the heightened contamination inputs from upstream sources and urban sewage discharge along the route", so the term about the water quality as relatively favorable is ambiguous.

Reviewer 3 ·

Basic reporting

Li and colleagues study focus in the bacterial community present along the urban artificial Jiusha River, and explored the variability of this bacterial community in association with human activities. The results show a clear distinction between the sampled points in the two major areas of the river, in respect to higher and lower pollution due to human activity

The manuscript is presented with clear language, appropriated literature and well referenced throughout the various sections.

The work is well explained and the manuscript correctly structured.

Figures and tables are relevant and appropriate. A point regarding figures should be checked below in the comments.

Unless I have missed anything during the downloading of the materials, I could not find any documents or links for the raw data, which should be made available.

Experimental design

Experimental design is clear and in general very well explained (see one point mentioned below in the comments).
Perhaps it would have been of interest for the conclusions and also for better understanding of this bacterial community, to include some extra sampling points in the confluence of the two major points, as well as after that area. It could provide good insights regarding the bacterial community when the more and less polluted areas merge as well as how that can affect the community on the areas after that confluence point.

Validity of the findings

The findings are very informative and well structured and explained.

Additional comments

- Make sure the raw data is available
- Introduction is clear and explains the state of the art and the main goal of the research project. Perhaps one short paragraph at the end could include the major finding of this research.
- Methods – in the sample collection, the authors should provide more detail about the method of collection and preparation/storage of sample until DNA extraction.
- In line 160, the acronyms ORP and DO appear without been described before. Make sure to either at this line or in the methods section make it clear for the reader that Oxidation reduction potential (ORP) and dissolved oxygen (DO). The same for EC in line 217.
- Figure 2B is misleading with the size of the circles. Line 197 the authors says that JHE4 and JHW1 had undetected Sulfuricurvum, but there is still a small dot in the plot. Either this is not correct, or the smallest dot means value 0. It would be great to have a scale for these circles to better interpret the numbers.
- It would be good to keep consistency in terms or colors amongst the figures. JHE is red but JHW is blue in some figures, but then green in figure 3.

---

## Round 0.2 · accepted · Accept

I agree with the reviewers that the manuscript is acceptable now.

Reviewer 2 ·

Basic reporting

I reviewed the corrected manuscript "Alteration of bacterial community composition in the sediments of an urban artificial river caused by sewage discharge" and the response letter that the authors have written. I agree with the authors' responses and the changes made to the manuscript, so I believe that the manuscript could be accepted for publication.

Experimental design

no comment

Validity of the findings

no comment

Additional comments

no comment

Reviewer 3 ·

Basic reporting

N/A

Experimental design

N/A

Validity of the findings

N/A

Additional comments

The authors carefully addressed the points and questions raised by the reviewers.
In my opinion this new corrected version is now suitable for publication.